# Structure and Dynamics of the Bacterial Flagellar Motor Complex

**DOI:** 10.3390/biom14121488

**Published:** 2024-11-22

**Authors:** Shuichi Nakamura, Tohru Minamino

**Affiliations:** 1Department of Applied Physics, Graduate School of Engineering, Tohoku University, 6-6-05 Aoba, Aoba-ku, Sendai 980-8579, Japan; shuichi.nakamura.e8@tohoku.ac.jp; 2Graduate School of Frontier Biosciences, Osaka University, 1-3 Yamadaoka, Suita Osaka 565-0871, Japan

**Keywords:** bacterial flagellum, chemotaxis, cryo-electron microscopy (cryo-EM), motility, proton motive force, rotor, stator, torque generation, transmembrane proton channel complex

## Abstract

Many bacteria swim in liquids and move over solid surfaces by rotating flagella. The bacterial flagellum is a supramolecular protein complex that is composed of about 30 different flagellar proteins ranging from a few to tens of thousands. Despite structural and functional diversities of the flagella among motile bacteria, the flagellum commonly consists of a membrane-embedded rotary motor fueled by an ion motive force across the cytoplasmic membrane, a universal joint, and a helical propeller that extends several micrometers beyond the cell surface. The flagellar motor consists of a rotor and several stator units, each of which acts as a transmembrane ion channel complex that converts the ion flux through the channel into the mechanical work required for force generation. The rotor ring complex is equipped with a reversible gear that is regulated by chemotactic signal transduction pathways. As a result, bacteria can move to more desirable locations in response to environmental changes. Recent high-resolution structural analyses of flagella using cryo-electron microscopy have provided deep insights into the assembly, rotation, and directional switching mechanisms of the flagellar motor complex. In this review article, we describe the current understanding of the structure and dynamics of the bacterial flagellum.

## 1. Introduction

The bacterial flagellum is a long filamentous organelle responsible for motility under various environmental conditions. In *Escherichia coli*, *Salmonella enterica* serovar Typhimurium (hereafter referred to as *Salmonella*), and other bacteria, the flagella are extracellularly located, with their proximal part embedded in the cell envelope [1]. On the other hand, the spirochetal flagella, called periplasmic flagella, are located within the periplasmic space. The periplasmic flagella serves as a cytoskeleton to maintain a spiral-shaped cell body [2,3]. Although the flagellar structure and its mechanical function are different among motile bacteria [4], the flagellum is generally divided into three functional parts: a basal body, a hook, and a filament. The membrane-embedded basal body acts as a rotary motor powered by a transmembrane electrochemical gradient of ions such as protons (H^+^) and sodium ions (Na^+^). The hook and filament extend into the cell exterior in *E. coli* and *Salmonella*. The long filament functions as a propeller. The flexible hook connects the basal body and filament and works as a universal joint that transfers the rotational force generated by the flagellar motor to the long helical filament (Figure 1) [5].

The flagellar motor of *E. coli* and *Salmonella* is composed of a rotor ring complex consisting of the MS-ring and the C-ring and a variable number of stator units. Because the flagellar motor of *E. coli* and *Salmonella* rotates counterclockwise (CCW; when viewed from the filament) and clockwise (CW), a reversible gear is incorporated into the C-ring structure [6,7,8].

The stator unit of the *E. coli* and *Salmonella* flagellar motor is formed by two transmembrane proteins called MotA and MotB. When the MotA-MotB complex assembles into the motor and is noncovalently anchored to the peptidoglycan (PG) layer through the well-conserved PG binding (PGB) domain of MotB, it can act as a transmembrane H^+^ channel that conducts H^+^ through the channel to generate torque by electrostatic interactions between MotA and FliG. The flagellar motor autonomously controls not only the number of stator units around the rotor but also the H^+^ channel activity of the MotA-MotB stator complex in response to changes in external load [6,9].

The direction of flagellar motor rotation is controlled by chemosensory systems. *E. coli* and *Salmonella* cells can sense chemical gradients in the environments, moving towards more favorable locations and away from less favorable ones for survival. Transmembrane chemoreceptors such as Tar and Tsr sense temporal changes in chemical stimuli in the environments and transmit such extracellular signals to the flagellar motor via a two-component signal transduction pathway. Transmembrane chemoreceptors regulate the autophosphorylation activity of the CheA kinase. Phosphorylated CheA transfers its phosphate group to a response regulator called CheY. Phosphorylated CheY (CheY-P) binds to the C-ring, allowing the motor to switch the direction of rotation from CCW to CW. The CheZ phosphatase removes the phosphate group from CheY-P. As a result, CheY dissociates from the C-ring, and the flagellar motor rotates CCW again [10,11].

Recently, the native hook-basal body structure isolated from *Salmonella* has been elucidated by high-resolution cryo-electron microscopy (cryo-EM) image analysis (Figure 1) [12]. Furthermore, the purified stator complex, which is composed of 5 MotA proteins and 2 MotB proteins, has been revealed at the atomic level by cryo-EM image analysis [13,14]. In this review article, we describe the structure and dynamics of the bacterial flagellar motor complex.

## 2. Flagellar Structure

The *Salmonella* flagellum consists of several ring structures called the MS-ring (FliF), the C-ring (FliG, FliM, FliN), the P-ring (FlgI), and the L-ring (FlgH), and an axial structure. The axial structure is divided into 5 parts: the rod (FliE, FlgB, FlgC, FlgF, FlgG), the hook (FlgE), the hook–filament junction (FlgK, FlgL), the filament (FliC or FljB) and the filament cap (FliD) (Figure 1). Except for the filament cap with 5-fold rotational symmetry, the axial structure is commonly a helical tubular structure with about 5.5 subunits per one turn of the helix. The N-terminal and C-terminal regions of the axial proteins form an α-helical coiled-coil structure, namely domain D0. Intermolecular interactions between the D0 domains are not only necessary for the assembly of the axial structure but also make it mechanically stable [15]. To build the axial structure beyond the cellular membranes, the axial component proteins pass through the cytoplasmic membrane via the flagellar type III secretion system (hereafter referred to as fT3SS), which is located inside the rotor ring complex, diffuse down a narrow channel in the growing structure, and assemble at the distal end [16,17].

### 2.1. MS-Ring

*Salmonella* FliF consists of 560 amino acid residues with two transmembrane helices (residues 26–46 and residues 455–457), and the 34 FliF proteins form the MS-ring within the cytoplasmic membrane [18,19]. The MS-ring serves not only as a part of the rotor of the flagellar motor but also as a structural template for flagellar assembly [20]. Furthermore, the MS-ring also houses the fT3SS export gate complex [21]. The periplasmic region of FliF contains three ring-building motifs: RBM1 (residues 60–124), RBM2 (residues 125–215), and RBM3 (residues 228–438). RBM3 consists of two domains: the S-ring domain (residues 228–270 and residues 382–438) and the β-collar domain (residues 271–381), which contains two sets of antiparallel β-sheets. The S-ring domains are horizontally packed with their major axis oriented in the radial direction to form the S-ring. The β-collar is a cylindrical β-barrel structure that extends above the S-ring. The structural flexibility of a loop located between RBM2 and RBM3 allows RBM2 to be oriented in two different directions relative to RBM3 within the MS-ring. As a result, 23 RBM2 domains of the 34 FliF proteins form the inner core of the M-ring, while the remaining 11 RBM2 domains, along with 11 RBM1 domains, form cog-like structures at the periphery of the ring (Figure 2a) [22,23,24,25,26]. The C-terminal cytoplasmic domain of FliF (FliF_C_, residues 514–560) interacts with the N-terminal domain of FliG, thereby connecting the MS-ring and C-ring [27,28,29,30,31].

### 2.2. C-Ring

The C-ring, formed by FliG, FliM, and FliN, has ~C34 symmetry [23,32,33]. The C-ring acts not only as a central part of the rotor for torque generation but also as a reversible gear that switches the direction of flagellar motor rotation [34].

*Salmonella* FliG (residues 1–331) consists of three globular domains called FliG_N_ (residues 1–67), FliG_M_ (residues 105–163), and FliG_C_ (residues 198–331). Two linker helices named Helix_NM_ (residues 68–104) and Helix_MC_ (residues 163–198) connect the FliG_N_ and FliG_M_ domains and the FliG_M_ and FliG_C_ domains, respectively (Figure 2b) [30,31,35]. FliG_N_ binds one-to-one to FliF_C_ [27,29], so the FliG-ring is formed by 34 FliG molecules on the cytoplasmic side of the MS-ring. Inter-subunit interactions between the FliG_N_ and FliG_N_ domains and between the FliG_M_ and FliG_C_ domains promote FliG-ring formation [36,37]. FliG_M_ binds one-to-one to FliM [38]. FliG_C_ is located at the top of the C-ring and is directly involved in the interaction with MotA for torque generation as well as for stator assembly around the rotor [39,40,41].

FliM (residues 1–331) and FliN (residues 1–137) form a cytoplasmic protein complex with a subunit stoichiometry of 1 FliM and 3 FliN [42], and this FliM_1_-FliN_3_ complex binds to the FliG-ring, forming the C-ring through the FliM_M_-FliG_M_ interaction (Figure 2b) [43]. FliM consists of a disordered N-terminal region (FliM_N_, residues 1–50) and two distinct globular domains called FliM_M_ (residues 51–233) and FliM_C_ (residues 234–331) [44,45], whereas FliN consists of a disordered N-terminal region (FliN_N_, residues 1–56) and a compactly folded C-terminal domain (FliN_C_, 56–137) that is structurally similar to FliM_C_ [46,47]. Inter-subunit FliM_M_-FliM_M_ interactions produce a continuous C-ring wall [45]. Together, FliM_N_ and FliN_C_ form a continuous spiral structure along the circumference at the lower part of the C-ring [30,31,48]. CheY-P binds to FliM_N_ and FliN_C_ in the C-ring, causing a highly cooperative transition of the C-ring structure from the CCW to the CW state [49,50,51].

The C-ring is essential for flagellar assembly because it provides the binding sites for the fT3SS ATPase ring complex [46,52,53]. So, if the C-ring is missing, flagellar assembly does not occur beyond the cytoplasmic membrane [54]. FliN_C_ alone supports flagellar protein export by the fT3SS, suggesting that FliN_N_ is dispensable for C-ring formation. In agreement with this, FliN_C_ alone can bind to the FliH_2_-FliI ATPase complex through an interaction between FliH and FliN; however, the removal of FliN_N_ from FliN considerably reduces flagella-driven motility but does not affect flagellar formation.This indicates that FliN_N_ is not required for flagellar protein export but is necessary for proper motor function [46,55].

### 2.3. LP-Ring

FlgH and FlgI are synthesized as precursor forms with cleavable N-terminal signal sequences, which are cleaved during the translocation across the cytoplasmic membrane via the Sec translocon [56]. FlgI is a periplasmic protein and assembles into the P-ring around the rod with the help of the FlgA chaperone [57,58]. The highly conserved Lys-63 and Lys-95 residues of FlgI are critical for the formation of the P-ring around the negatively charged surface of the rod [59]. The lipoprotein, named FlgH, assembles into the L-ring within the outer membrane [60]. The N-terminal extended chain of FlgH binds to two hydrophobic pockets of FlgI, so the L- and P-rings together form the LP-ring complex with 26-fold rotational symmetry (Figure 2c). The N-terminal cysteine residue in the N-terminal extended chain of FlgH is modified by fatty acylation that allows the LP-ring to anchor to the outer membrane through interactions with lipopolysaccharide. The fatty acyl group attached to this cysteine residue is inserted into the hydrophobic gap between the next two FlgH subunits, thereby forming a hydrophobic band around the L-ring with the lipopolysaccharide layer. In the LP-ring structure, each subunit makes complex intermolecular interactions with up to six subunits, making it mechanically stable [25,26,59]. Because the P-ring firmly associates with the rigid PG layer, the LP-ring complex serves as a bushing for the rod, which acts as the drive shaft of the flagellar motor, to rotate at high speed. Because the outer surface of the rod and the inner surface of the LP-ring are very smooth, the rod can rotate freely without friction within the LP-ring [59]. This is supported by experimental data showing that the flagellar motor without the functional stator units exhibits rotational diffusion due to Brownian motion [61]. There are both negatively and positively charged residues on the inner surface of the LP-ring complex. In contrast, the outer surface of the rod is strongly negatively charged by three negatively charged residues in FlgG, Asp-109, Asp-154, and Glu-203. These findings suggest that the rod can rotate in the center of the LP-ring due to the action of both repulsive and attractive forces [59].

Because of the difference in the cell envelope of Gram-negative and Gram-positive bacteria, the LP-ring complex is missing in the basal body of Gram-positive bacteria such as *Bacillus subtilis* [62]. The PG layer of *B. subtilis* is much thicker than that of *Salmonella*, but the rod of the *B. subtilis* flagellar motor passes completely through the PG layer. Because the thick PG layer is a very rigid structure, it is believed to serve as a bushing for the rod to rotate at high speed. The periplasmic flagella do not have the L-ring [2,3]. Thus, the LP-ring complex is not a highly conserved flagellar structure among motile bacterial species.

### 2.4. Rod

The rod is a helical structure consisting of 11 protofilaments [63]. The rod is subdivided into two parts: the proximal rod is composed of 6 FliE subunits, 5 FlgB subunits, 6 FlgC subunits, and 5 FlgF subunits, and the distal rod is formed by 24 FlgG subunits [25,26,64]. The proximal rod is in the β-collar of the MS-ring, and the distal rod is in the LP-ring complex. FliE has three α-helices with the α2 and α3 helices forming domain D0. FlgB, FlgC, FlgF, and FlgG contain the D0 and Dc domains. FlgF and FlgG also contain the D1 domain (Figure 3a). The Dc domain adopts a long β-hairpin structure, named L-stretch, making the rod straight and stiff to serve as a drive shaft of the flagellar motor [65,66]. Rod assembly begins with the FliE zone at the tip of FliP and FliR in the fT3SS, followed by the FlgB, FlgC, and FlgF zones in this order, and finally, FlgG assembles into the distal rod. The N-terminal α1 helix of each FliE subunit binds to the inner wall of the MS-ring, firmly connecting the MS-ring and the rod. Thus, FliE prevents the rod from dislodging from the MS-ring as the flagellar motor rotates under a variety of environmental conditions [67].

### 2.5. Hook

The hook is flexible for bending but rigid against twisting, and this bending flexibility is important for the hook to function as a universal joint [68]. The hook is a helical tubular structure composed of 11 protofilaments [69]. *Salmonella* FlgE consists of domains D0, Dc, D1, and D2, arranged from the inner to outer parts of the hook structure (Figure 3a) [70,71]. The D0, Dc, and D1 domains of FlgE show structural similarities to those of FlgG, allowing a direct connection between the distal rod and hook (Figure 3b) [63,72]. The length of the L-stretch of FlgE is 18 amino acid residues shorter than that of FlgG, creating gaps between FlgE subunits in the hook structure. These gaps allow the curved hook structure to compress and extend sequentially during high-speed flagellar rotation [73].

The hook of *Salmonella* is formed by about 120 FlgE subunits, so the length of the hook is fairly well controlled at about 55 nm [74]. The hook length control is also important for the proper universal joint function of the hook [75,76]. The hook length is measured by the FliK ruler protein secreted by the fT3SS during hook assembly [77,78,79]. When the hook reaches its mature length of about 55 nm, FliK transmits the hook length signal to the fT3SS through the interactions with two fT3SS proteins called FlhB and FlhA, not only terminating hook assembly but also initiating filament assembly at the hook tip [80,81]; therefore, *Salmonella* mutant strains with loss-of-function of FliK or specific amino acid substitutions in FlhA or FlhB produce unusually elongated hooks called polyhooks that lack the filament structure.

The purified polyhook structure undergoes polymorphic transformations depending on pH, temperature, and salt concentration [82]. High-resolution cryo-EM structural analysis of the curved hook structure has revealed that the conformation of the FlgE subunit changes gradually from one protofilament to the next. The conformation of FlgE is essentially the same within each protofilament, so there are 11 distinct conformations of FlgE in the curved hook structure. Clearly, differences in the close inter-subunit interactions between the D2 domains are apparently responsible for the differences in the curved hook structures. This suggests that the curvature and twist of each supercoil depend on the direction of the intermolecular D2-D2 interactions [83,84]. In agreement with this, the deletion of domain D2 from FlgE makes the hook straight [85].

### 2.6. Filament

Flagellin is the building block of the flagellar filament. The flagellar filament is the largest part of the flagellum and grows to a length of about 15 μm by the assembly of ~30,000 flagellin subunits with the help of the filament cap formed by five FliD subunits [86,87,88,89,90]. The filament is a helical tubular structure composed of 11 protofilaments, as in the rod and hook [91]. Because of differences in the structure and function between the hook and filament (Figure 3a), the newly transported flagellin subunit cannot assemble directly at the hook tip; therefore, the hook–filament junction structure, formed by 11 FlgK subunits and 11 FlgL subunits, functions as a structural adapter that firmly connects the hook and filament [92,93,94,95]. If FlgK, FlgL, or FliD is missing, flagellin subunits transported from the cytoplasm by the fT3SS cannot polymerize into the filament and simply leak out into the culture media [96].

*E. coli* and *Salmonella* produce peritrichous flagella (typically about 5 to 10 per cell) that are randomly arranged on the cell surface. This flagellar filament undergoes polymorphic transformations of its supercoiled form during motility [97,98]. Interestingly, the polar flagellar filament from *Pseudomonas aeruginosa* undergoes polymorphic transformation, whereas that from *Vibrio parahaemolyticus* and *Caulobacter crescentus* does not [99].

The left-handed supercoil of the *Salmonella* filament is named “normal”. When all flagellar motors rotate CCW, several normal filaments come together to form a flagellar bundle structure behind the cell body due to the bending flexibility of the hook, enabling the *Salmonella* cell to swim straight. The switch in the direction of the motor from CCW to CW produces the mechanical force, which converts the normal filament to right-handed helical forms named “semi-coiled”, “curly I”, or “curly II”. As a result, the bundle is disrupted, enabling the cell to tumble and change the swimming direction [97,98].

Polymorphic transformation of the filament structure has been explained by a bi-stable protofilament model [100,101,102,103]. Each of the 11 protofilaments can be in one of two conformational states: left-handed (L-type) and right-handed (R-type). These two different protofilaments exhibit different lengths in axial periodicity and different helical lattices, left-handed and right-handed, in lateral interaction; therefore, the helical properties of each supercoil are determined by a ratio of L-type protofilaments to R-type ones in the filament structure; however, recent high-resolution cryo-EM image analyses of the normal flagellar filament with a pitch of 2.3 μm and a diameter of 0.4 μm and the curly I filament with a pitch of 1.1 μm and a diameter of 0.3 μm have revealed that in these two supercoiled filaments, the conformation of the flagellin subunit gradually changes from one protofilament to the next, indicating that the supercoiled structure of the flagellar filament is created by 11 distinct conformations of flagellin in the filament. The 11 conformations of the protofilaments also vary with the curvature of the supercoiled form so that the protofilaments are more similar to each other in the normal filament and more different in the curly I filament. This is because the degree to which each protofilament bends into a curve is greater in the curly I state than in the normal state. The major differences in protofilament and subunit conformations between the normal and curly I filament structures result not only from the curvature but also from the need to maintain the central channel inside the filament that is a physical path for newly transported flagellin subunits to diffuse to the growing end of the filament structure. These findings would cast doubt on the bi-stable protofilament model [104].

Flagellin is known as the H-antigen because it is a major target of the host immune system. *Salmonella* flagellin is composed of domains D0, D1, D2, and D3, arranged from the inner to the outer part of the filament structure (Figure 3a) [105,106]. Domains D0 and D1 are well conserved, whereas domains D2 and D3 are variable even among *Salmonella* species because these two domains are the major targets of antibodies [107]. *Salmonella* has two different flagellin genes named *fliC* and *fljB* on the genome, and their expression is stochastically switched at a frequency of 10^−3^–10^−5^ per cell per generation. As a result, each *Salmonella* cell produces either FliC or FljB filaments. This phenomenon is called flagellin phase variation. This alternating expression of these two different flagellin genes allows *Salmonella* cells to escape from the host immune system by altering flagellar antigen specificity [108].

The swimming motility of *Salmonella* cells expressing only FliC is worse under high viscous conditions than that of the cells expressing only FljB. Cryo-EM structural analysis of the FliC and FljB filaments has shown that the overall structure of FljB is almost the same as that of FliC, but the position and orientation of domain D3 of FljB are different from those of FliC (Figure 3c). Furthermore, this D3 domain is more flexible and mobile in the FljB filament than in the FliC filament. These findings suggest that these structural differences in the D3 domain of FliC and FljB cause functional differences between the FliC and FljB filaments; therefore, the flagellin phase variation may also contribute to optimizing the propeller function of the filament under different environments [109,110,111].

### 2.7. FT3SS

The fT3SS consists of a transmembrane export gate complex and a cytoplasmic ATPase ring complex (Figure 1). The export gate complex, which consists of 9 FlhA molecules, 1 FlhB molecule, 5 FliP molecules, 4 FliQ molecules, and 1 FliR molecule, is located within the central pore of the MS-ring and uses the proton motive force (PMF) across the cytoplasmic membrane to drive H^+^-coupled protein export [112,113]. The cytoplasmic ATPase complex, which is made up of 6 FliH homo-dimers, 6 FliI molecules, and 1 FliJ molecule, associates with the C-ring through an interaction between FliH and FliN and acts as an ATP-driven activator for the export gate complex to become a H^+^-driven protein transporter [114]. The cytoplasmic FliH_2_-FliI complex not only serves as a dynamic carrier that escorts chaperone-associated export substrates from the cytoplasm to the docking platform formed by the C-terminal cytoplasmic domains of FlhA and FlhB [115,116] but also helps the docking platform achieve the proper order of flagellar protein export in parallel with the assembly order of the flagellar axial structure [117].

FliP and FliR assemble into the FliP_5_-FliR_1_ complex, and 4 FliQ subunits then bind to the FliP_5_-FliR_1_ complex to form the FliP_5_-FliQ_4_-FliR_1_ complex that allows proteins of the axial structure of the flagellum to pass through the cytoplasmic membrane [118]. Because the FliP_5_-FliQ_4_-FliR_1_ complex has a helical symmetry inside the MS-ring, domain D0 of FliE directly assembles on the tip of FliP and FliR to form the most proximal part of the rod [25,26]. FlhB associates with the FliP_5_-FliQ_4_-FliR_1_ complex and plays an important role in opening and closing the cytoplasmic gate of the polypeptide channel [119,120]. FlhA forms a nonameric ring structure around the FliP_5_-FliQ_4_-FliR_1_-FlhB_1_ complex [121,122]. The FlhA_9_ ring couples inward-directed H^+^ flow through the FlhA channel with outward-directed protein translocation through the polypeptide channel [123].

## 3. Stator Assembly

The flagellar rotor ring complex is surrounded by a dozen stator units, and electrostatic interactions between MotA and FliG drive flagellar motor rotation. To directly observe the stator assembly process, Block et al. used an *E.coli motA*-*motB* deletion mutant strain transformed with a plasmid encoding both *motA* and *motB*, induced the expression of the MotA and MotB stator proteins from the plasmid, and observed stepwise increments in the rotation rate of a single *E. coli* cell tethered to a cover slip [124]. This speed recovery, called “resurrection”, indicates that each stator unit is assembled to the motor independently. Speed fluctuations are also observed during flagellar motor rotation. These fluctuations are caused by variations in the number of functional stator units in the motor. Using high-resolution single-molecule imaging of MotB labeled with a green fluorescent protein (GFP), Leake et al. have provided direct evidence that the stator unit shows rapid exchanges between the motor and the membrane pool while the motor is rotating [125]. Thus, the flagellar stator units are not always assembled to the motor but dissociate frequently, even during rotation (Figure 4a). Since then, factors controlling stator dynamics have been explored.

### 3.1. Factors Controlling Stator Assembly

#### 3.1.1. Coupling Ions

The coupling ions such as H^+^ and Na^+^ not only drive the motor rotation but also induce the assembly of the stator unit into the flagellar motor. A typical example is a marine bacterium, *Vibrio alginolyticus*, which needs environmental (external) Na^+^ for the incorporation of the Na^+^-type PomA-PomB stator complex into a polar flagellar motor [126]. In *E. coli*, the assembly of the MotA-MotB complex into the flagellar motor depends on the PMF across the cytoplasmic membrane, so PMF disruption induces the dissociation of functional stator units from the motor [127,128]. In contrast to the *E. coli* flagellar motor, the dissociation of the MotA-MotB complex does not occur in the *Salmonella* flagellar motor when the PMF is significantly reduced [41]; however, as the external pH decreases, more stator units are incorporated into the motor compared to neutral pH [129]. In addition, a loss of the H^+^ channel activity of the MotA-MotB stator complex (e.g., due to the replacement of Asp-33 to Asn in *Salmonella* MotB) significantly reduces the binding affinity of the MotA-MotB complex for the flagellar motor in *Salmonella* [130]. Because each stator unit is stabilized around the rotor via a tight association between the PGB domain of MotB and the PG layer, the dependence of stator assembly on the coupling ion probably involves ion-dependent structural changes in the stator unit that opens the ion channel [131,132].

#### 3.1.2. Load

External load on the motor affects the stability of the stator assembly. Tipping et al. varied the external load by changing the size of beads attached to the flagellar filament and the viscosity of the medium and showed that the fluorescence brightness of the functional stator units labeled with GFP increases with increasing the external load [133]. Lele et al. successfully captured motor rotation immediately after attaching a bead to a stub of a “sticky” flagellar filament and showed that abrupt increases in load cause stepwise increases in motor speed [134]. This result suggests that a few stator units spin the motor at a low load and that the number of active stator units increases with an increase in external load. Thus, the flagellar motor can autonomously control the number of functional stator units around the rotor in response to changes in external load. When the flagellar motor operates at a very low load, close to zero, there is typically only one stator unit around the rotor; therefore, long pauses in motor rotation are frequently observed. Interestingly, increasing the external load or overexpression of the MotA-MotB stator complex suppresses these pausing events, suggesting that the bound lifetime of each stator unit incorporated into the motor becomes much shorter when the motor operates near zero loads [61].

The mechanism of load-dependent stability of the stator assembly has not been elucidated, but Nord et al. have explained it with a “catch-bond” mechanism (Figure 4b) [135]. Various biological interactions are enhanced by pulling forces, shear flows, and other external forces, as seen in cell adhesions and linear molecular motors. Nord et al. manipulated the load on the flagellar motor using magnetic microbeads placed in a magnetic field and observed the dynamics of the stator unit upon the repeated stall and release (i.e., load elevation and reduction) of motor rotation. These load-controlled experiments have shown that the flagellar rotor, free from magnetic constraints, is rotated by ~5 stator units, and the trap-dependent stall allows several more stator units to be incorporated into the motor.

Measurements of stator kinetics have revealed that the dissociation rate of the stator unit ($k_{off}$) depends on the external load and the local force exerted on the rotor by the individual stator units, which is estimated from the average torque produced by a single stator unit and the rotor radius, whereas the association rate of the stator unit ($k_{on}$) does not. Greater force per stator unit alters the conformation of the PGB domain of the stator unit fixed to the PG layer, which retards the dissociation of the stator unit from the rotor and extends the bound lifetime of each stator unit. As the external load increases, the binding affinity of CheY-P for the flagellar motor increases, switching the direction of motor rotation from CCW to CW even at low CheY-P concentrations [136]. Thus, autonomous load-dependent stator remodeling is important in fine-tuning not only motor speed but also chemotaxis.

#### 3.1.3. Number of Assembled Stator Units in the Flagellar Motor

Wadhwa et al. have proposed a model in which stator binding is promoted as motor speed increases, resulting in an increase in the number of functional stator units around a rotor [137]. Ito et al. have shown that the probability of assembling a new stator unit into a non-rotating motor without functional stator units is significantly lower than the probability of assembling it into a motor rotated by previously assembled stator units (Figure 4c) [138]. In other words, the new stator unit can be easily placed around the rotor while the already assembled stator units spin the rotor. According to the catch-bond mechanism, the pulling force generated between the assembled stator units and the rotating rotor strengthens the stator anchoring to the PG layer. An adequate force generation to control stator activation may involve a gear-like interaction between several stator units and the rotor, as discussed later. Wadhwa et al. have also observed an increase in the stator binding rate from non-rotation to rotation states, but the binding rate peaked at an intermediate number of installed stator units [139]. They have explained the phenomenon using two counteracting kinetic parameters that are functions of the number of stator units: One of the parameters is that at increased motor speed, there is an increased probability of collision between an unbound stator and the rotor; the other is that there is a decreased probability of forming a permanent binding of a stator unit because of its shorter contact time with the more-rapidly spinning rotor.

**Figure 4 biomolecules-14-01488-f004:**
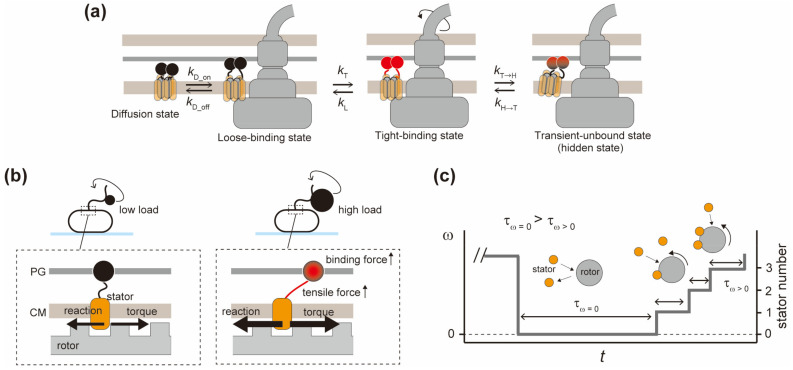
Stator activation and assembly. (**a**) A four-state model for stator assembly. The proton channel is unplugged in the transition from the diffusion state to the loose-binding state, as shown in (**b**). Wadhwa et al. assumed that the transition between the loose-binding and tight-binding states depends on torque [139]: $k_{T}/k_{L}$ increases with torque in agreement with (**b**). In the transient-unbound state, the stator unit is hypothesized not to produce torque but to stay in the vicinity of the rotor and return to the bound, torque-producing state for a very short time: $k_{H\to T}\gg k_{T\to H}$. PG, peptidoglycan layer; CM, cytoplasmic membrane. (**b**) Catch-bond mechanism. (**c**) Stator binding probability depends on motor rotation. The time for stator assembly to a rotating motor ($\tau_{\omega >0}$) is shorter than that of a non-rotating motor where no stator is docked ($\tau_{\omega =0}$).

#### 3.1.4. FliL

*E. coli* and *Salmonella* can migrate over the surface of the semi-solid medium by means of flagella. This flagella-driven surface motility is called “swarming”. FliL is a stator-associated protein essential for swarming on surfaces but not for swimming in liquids [140], suggesting that FliL modulates the function of the stator unit. FliL is a single-transmembrane protein with a large C-terminal periplasmic domain (FliL_C_). Interestingly, FliL_C_ shows a remarkable structural similarity to stomatin family proteins, which control the ion channel activity in various organisms. FliL_C_ forms a decameric ring, and the MotA-MotB complex is located within the central pore of the FliL_C_ ring [141,142]. The linker region between the N-terminal transmembrane helix and the C-terminal PGB domain of MotB interacts with the FliL_C_ ring [143]. Because this linker region regulates the load-dependent stator assembly dynamics [144], the FliL ring stabilizes the active conformation of the MotA-MotB complex, allowing it to work as a stator unit in the flagellar motor efficiently.

### 3.2. Multi-State of Stators

Two distinct states of the stator unit, bound and unbound, have been assumed; however, actual stator assembly dynamics are more complicated, and some experiments have predicted multiple states of the stator unit. Shi et al. have observed a double-exponential decay of the stator-exchange intervals, proposing two types of transition, between unbound and bound states with a slower rate constant and between bound and hidden states with a faster rate constant [145]. Wadhwa et al. have measured stepwise stator remodeling and have realized that the time distribution of step intervals (time until the next stator binding) could not be reproduced by a two-state kinetic model assuming unbound and bound states (Figure 4a) [139].

The kinetics of stator remodeling is best described by a four-state mechanism. Noting the load dependence of the stator-docking lifetime, there is a loose-binding state, in which the stretch of the stator anchoring region is limited because of low torque, and a tight-binding state, where the stator anchoring region stretches in response to high torque. There are also two unbound states, one being free diffusion in the membrane and the other being the hidden state proposed by Shi et al., in which the stator unit is still bound to PG at the periphery of the rotor but is not engaged with FliG (Figure 4a).

## 4. Motor Rotation Mechanism

### 4.1. A Traditional Rotation Model

The central event in the flagellar rotation is stator-rotor interaction coupled with the translocation of ions through the ion channel in the stator complex. In the longstanding mainstream model of the H^+^-driven motor of *E. coli*, the stator unit was believed to be formed by 4 MotA subunits and 2 MotB subunits. Because MotB forms a homo-dimer, the stator complex contains two distinct H^+^ pathways for proton translocation, and a highly conserved Asp-32 residue in *E. coli* MotB (Asp-33 in *Salmonella* MotB) is a crucial H^+^-binding site. The protonation and deprotonation cycle of MotB-Asp32 induces cyclic conformational changes in the cytoplasmic domain of MotA, which is followed by the electrostatic MotA-FliG interactions via pairs of charged residues, such as MotA-R90, MotA-E98, FliG-D281, FliG-D288, and FliG-D289 [146]. Based on this mechanism, theorists have proposed kinetic or stochastic models to explain the torque-speed curve [147], stepwise rotation [148], motor reversal [149], and the relationship between stator remodeling and motor speed [150,151].

### 4.2. The Latest Rotation Model

Recently, high-resolution cryo-EM image analysis has solved several structures of the stator complexes from different bacterial species and the CCW and CW forms of the rotor ring complex at near-atomic resolution. These structures have completely revised models for the mechanism of rotor rotation. Santiveri et al. have obtained an almost complete atomic model of MotA and MotB derived from *Campylobacter jejuni* and have shown that the stoichiometry of MotA:MotB is 5:2 (Figure 5a) [152]. They have also revealed that the MotA-MotB complex of *Shewanella oneidensis* and the PomA-PomB complex of *V. alginolyticus* are composed of 5 A subunits and 2 B subunits. Moreover, Deme et al. have reported the cryo-EM structures of the MotA_5_-MotB_2_ complex derived from *Vibrio mimicus*, *Clostridium sporogenes*, and *B. subtilis* [153]. Thus, 5 A:2 B is the common stoichiometry in the bacterial flagellar stator complex.

The five MotA proteins assemble into a homo-pentameric ring structure (MotA_5_), and two transmembrane helices of MotB (MotB_TMH_) form an α-helical coiled-coil that penetrates the central pore of the MotA_5_ ring. Limited proteolysis assays showed that the mutation causing the D32N residue substitution in *E. coli* MotB, which mimics the protonation of MotB-Asp32, causes a large conformational change in the cytoplasmic domain of MotA (MotAc) [154]. The authors proposed that this change reflects a power-stroke mechanism for the stator. In contrast, the cryo-EM structure of the MotA_5_-MotB_2_ ring complex from *C. jejuni* reveals that the D22A change does not induce a large conformational change in MotAc [152](Asp-22 in *C. jejuni* corresponds to Asp-32 in *E. coli*).

The following estimate suggests the need for rotation of the MotA_5_ ring, which is a wheel, relative to the MotB_TMH_, which is an axle [152]. The rotor ring, with a diameter of ~450 Å and a circumference of ~1420 Å, is formed by 34 FliG subunits [23]; therefore, adjacent FliG subunits are about 1420/34 = ~42 Å arc length apart. That should be the elementary step length of the motor. A root-mean-square-deviation (RMSD) scoring the structural difference) in Cα atoms between the deprotonated and protonated states of the MotA_5_-MotB_2_ ring complex is only 0.297 Å. Thus, there must be very considerable extra movement of MotAc coupled with H^+^ translocation through the ion channel.

The diameter of the MotA_5_-MotB_2_ ring structure is ~75 Å, and the circumference is ~230 Å. In a 72° rotation of the MotA ring around the MotB dimer, it will move through an arc of ~46 Å. Thus, the influx of H^+^ through the H^+^ channel induces rotation of the MotA_5_ ring of the proper size, given the inaccuracy of the precise measurements of the distance of just the right magnitude to propel a 42 Å rotation of the FliG rotor ring. Similar H^+^-coupled structural changes in the stator complex are also observed in the spirochete *Borrelia burgdorferi* [155]. The rotational switching mechanism proposed for the *B. burgdorferi* flagellar motor employs a similar mechanism of stator rotation (discussed later).

## 5. Switching

The C-terminal domains of the MotA_5_ ring interact with FliG_C_ in the C-ring to power flagellar motor rotation. The directional switching of flagellar rotation is regulated by the chemotaxis signaling pathway. When CheY-P binds to FliM and FliN in a C-ring, the conformational change in FliG_C_ occurs, resulting in a change in the physical interaction faces between FliG and MotA and a switch in the direction of flagellar motor rotation from CCW to CW.

FliG_C_ is divided into two subdomains: FliG_CN_ and FliG_CC_. A highly conserved MFXF motif connects these two subdomains. Structural insights into the FliG conformational change have been proposed based on different crystal structures of the FliG subunit [35,156,157]. The motor rotation is reversed by a 180° rotation of the FliGcc subdomain relative to the FliG_CN_ subdomain through a conformational change in the conserved MFXF motif. This idea is supported by recent cryo-EM structures of the C-rings in the CCW and CW states. Furthermore, the cryo-EM structures also show that FliG_N_ rotates 180° relative to FliG_M_ upon binding of CheY-P to the C-ring. These conformational changes in the FliG-ring seem to be critical for the directional switching of flagellar motor rotation [30,31,48].

The MotA_5_ ring is expected to only rotate in the CW direction due to steric hindrance between the MotA_5_ ring and MotB_TMH_. Based on the stator rotation mechanism, recent rotational switching models emphasize a mechanism whereby bidirectional rotation of the rotor ring is achieved by unidirectional rotation of the stator unit driven by inward-directed H^+^ movement through two distinct H^+^ translocation pathways in the MotA_5_-MotB_2_ complex. Chang et al. have analyzed the *B. burgdorferi* flagellar motor with or without CheY-P bound to the C-ring using cryo-electron tomography and have shown that interactions of CheY-P with FliM and FliN in the C-ring induce a large conformational change in FliG2 (major FliG subunit in *B. burgdorferi*), thereby increasing the diameter of the FliG2-ring [158]. The stator units are fully assembled into the *B. burgdorferi* flagellar motor in vivo. These stators interact with the outside of the FliG2-ring in the absence of CheY-P (CCW), whereas they interact with the inside of the ring in the presence of CheY-P (CW). The change in the interaction site between the stator unit and the FliG2-ring can explain bidirectional motor rotation by unidirectional stator rotation.

Johnson et al. used *C. sporogenes* to analyze the structure of the MotA-FliG_CC_ fusion formed via an unstructured linker to understand the intermolecular interaction between MotA_C_ and FliG_CC_ (Figure 5b) [30]. When the fusion structure is superimposed onto the C-ring in the CCW and CW states, focusing on the MotA-FliG_CC_ interface, the binding of CheY-P shifts the binding site of each stator unit to the FliG-ring from the outside to the inside, as proposed in *B. burgdorferi*.

## 6. Conclusions and Perspectives

This review article describes recent findings on the structure and dynamics of the bacterial flagellar motor complex. High-resolution cryo-EM image analysis is one of the techniques that has contributed most to unraveling various structural mysteries, such as the symmetry mismatch seen in the flagellar structure, the functional mechanism of the LP-ring complex acting as the molecular bushing, and a supercoiling mechanism of the hook and filament structures. Moreover, structural analysis by cryo-EM has revealed the structure of the stator complex, proposing a plausible model for torque generation in which the MotA_5_ ring rotates around the MotB_THM_ in response to the influx of H^+^ through the H^+^ channel. Experiments and data analyses in the context of physics have provided insight into the stator assembly mechanism.

Thus, although our understanding of the bacterial flagellum has advanced in the past few years, single-molecule measurement experiments aimed at elucidating the flagellar rotation mechanism, which was once very active, seem to have become somewhat less common. The stator-rotation model provided by cryo-EM studies requires researchers to capture somehow and quantify the rotation of the stator in the functional flagellar motor. ATP-driven molecular motors that are successfully extracted from cells, such as actomyosin, kinesin-dynein, and F_1_-ATPase, have been analyzed in vitro, allowing precise input energy control and motivating theorists to model the operation mechanisms of the ATP-driven motors. In vitro assay of the bacterial flagellum was established in the fT3SS [159] but has not yet been applied to rotational experiments of the flagellar motor reconstituted in vitro. (Reconstitution of the PG layer in membrane vesicles is required for anchoring the stator complexes). High-resolution dynamic measurements of the flagellar motor complex, adopting in vitro measurements if possible, will provide a more in-depth understanding of the torque generation mechanism of the flagellar motor.

## Figures and Tables

**Figure 1 biomolecules-14-01488-f001:**
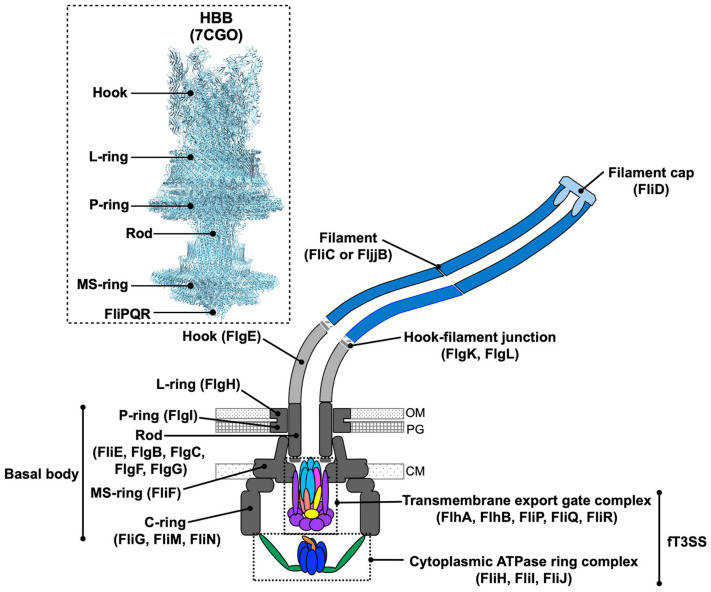
Schematic diagram of the *Salmonella* flagellum. The flagellum consists of a type III secretion system (fT3SS), a membrane-embedded basal body, a short flexible hook, a hook–filament junction, a long filament, and a filament cap. Several stator units, formed by 5 MotA subunits and 2 MotB subunits, surround the basal body, although they are not shown in this diagram. The inset shows the cryo-EM structure of the hook-basal body isolated from *Salmonella* (PDB ID: 7CGO). OM, outer membrane; PG, peptidoglycan layer; CM, cytoplasmic membrane.

**Figure 2 biomolecules-14-01488-f002:**
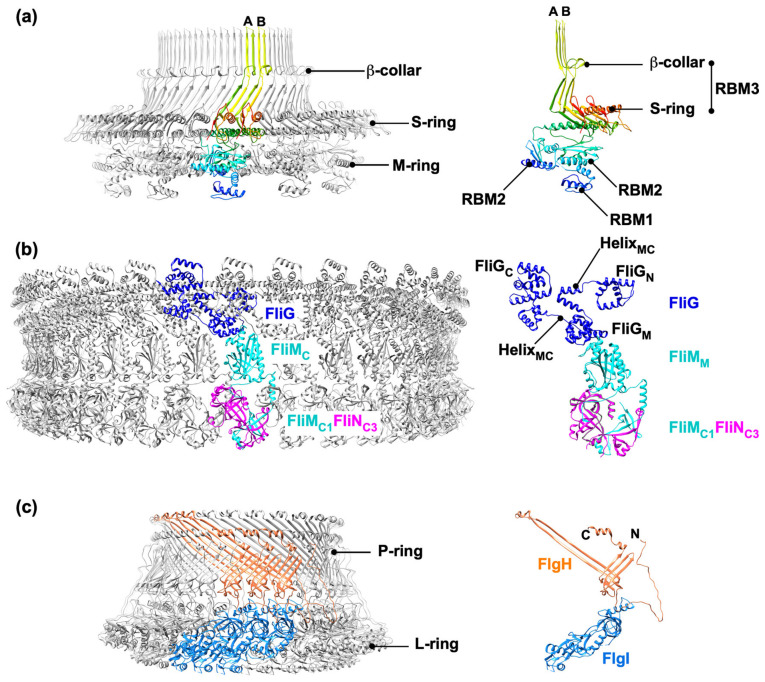
Cryo-EM structures of ring complexes in the *Salmonella* basal body. (**a**) Cα ribbon diagram of the atomic model of the MS-ring (PDB ID: 8T8P). Only two Cα backbones (Mol-A and Mol-B) are color-coded from blue to red, going through the rainbow colors from the N-terminus to the C-terminus. FliF has three ring-building motifs: RBM1, RBM2, and RBM3. RBM3 forms the S-ring and β-collar with C34 symmetry. FliF has two different conformations in the MS-ring, so RBM2 in Mol-A faces inward, and in Mol-B faces outward. RBM1 is missing in Mol-A. As a result, 23 RBM2 domains face inward to form the inner core ring of the M-ring, while the remaining 11 RBM1-RBM2 domains form cog-like structures just outside the inner core ring. (**b**) Atomic models of the C-ring (PDB ID: 8UOX) and the FliG_1_-FliM_1_-FliN_3_ complex (PDB ID: 8UMD) in Cα ribbon representation. The C-ring is composed of 34 FliG subunits, 34 FliM subunits, and 102 FliN subunits. FliG (blue) consists of three domains, FliG_N_, FliG_M_, and FliG_C_ and two helix linkers named Helix_NM_ and Helix_MC_. FliM (cyan) consists of an intrinsically disordered N-terminal region and two compactly folded domains, FliM_M_ and FliM_C_. FliN (magenta) consists of an intrinsically disordered N-terminal region and a compactly folded domain, FliN_C_. FliM_M_ binds to FliG_M,_ whereas FliM_C_ forms a spiral structure along with the three FliN_C_ domains. (**c**) Cα ribbon diagram of the atomic model of the LP-ring complex (PDB ID: 8WHT). The LP-ring complex is composed of 26 FlgH subunits and 26 FlgI subunits. The three FlgH subunits are colored in coral, whereas the three FlgI subunits are colored in dodger blue. The N-terminal disordered chain of FlgI binds to FlgH.

**Figure 3 biomolecules-14-01488-f003:**
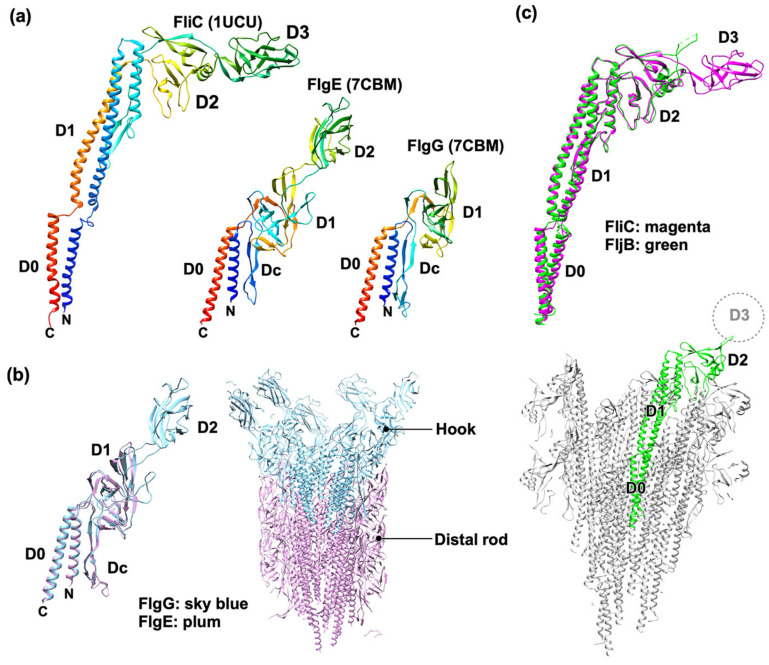
Atomic models of flagellar axial proteins. (**a**) Cα ribbon diagrams of the distal rod protein (FlgG) (PDB ID: 7CBM), the hook protein (FlgE) (PDB ID: 7CBM), and the filament protein (FliC), namely flagellin (PDB ID:1UCU) derived from *Salmonella*. The N-terminal and C-terminal α-helices form an α-helical coiled-coil structure (domain D0). (**b**) Structural comparison of FlgE (plum) and FlgG (sky blue). Domains D0, Dc, and D1 of FlgE are structurally similar to those domains in FlgG (left panel), so newly transported FlgE molecule assembles into the hook at the tip of the distal rod (right panel). (**c**) Structural comparison of the R-type FliC subunit (PDB ID: 1UCU, magenta) and the R-type FljB subunit (PDB ID: 6JY0, green). Domains D0, D1, and D2 of FljB are nearly identical to those of FliC (top panel); however, the electron density corresponding to domain D3 of FliC is very poor in the FljB filament structure (lower panel); the cryo-EM structure of this domain is missing.

**Figure 5 biomolecules-14-01488-f005:**
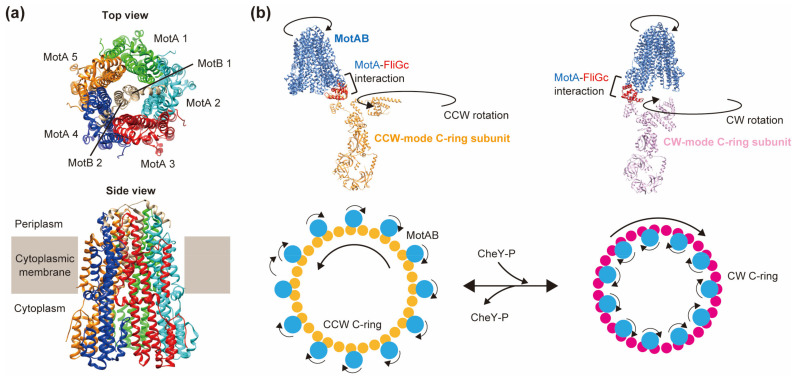
Structure of the stator complex and bi-directional rotation model. (**a**) MotA_5_-MotB_2_ complex of *Campylobacter jejuni* (PDB ID: 6YKM). (**b**) The structure of *Clostridium sporogenes* MotA–FliG_C_ fusion (PDB ID: 8UCS) overlaid on the FliG_C_ domain of the CCW (PDB ID: 8UMD) (upper left) and CW C-ring structures (PDB ID: 8UMX) (upper right) rotor subunits. A model for CCW/CW bidirectional rotor rotation based on unidirectional CW stator rotation (viewed from the cytoplasmic membrane looking down onto the FliG ring; lower panels).

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
