# Peer review of "Structure and Dynamics of the Bacterial Flagellar Motor Complex"

_biomolecules, 2024, doi:10.3390/biom14121488_

Round 1
Reviewer 1 Report
Comments and Suggestions for Authors
This is a timely, concise, and extremely well-written review. I have spent a lot of time making minor editorial suggestions, almost all of which reflect the fact that the authors, despite their excellent writing, are not native English speakers and thus get a few things, like the use of articles and punctuation, wrong. I have also rewritten a few sentences and paragraphs to make them, in my opinion, clearer to the reader.
I have only one major criticism, and it is easily dealt with. Figure 4a is inaccurate and misleading. It suggests that the stator complex takes on the conformation in which its ion channels are open and its PGB domain is properly folded before it contacts the FliG ring. I do not think that is correct. Is there any evidence that the coupling ion binds directly to the PGB domain of the MotB? I am unaware of it, and it seems unlikely, as those ions are always present.
My understanding, and I have thought about this a lot, is that contact of the stator unit with FliG causes the plugs on the ion channels to be removed and opens the ion channel and initiates assembly of the PGB domain into a conformation that can bind PG. FliG does not enter into the mechanism shown in Fig. 4a, which seems like a serious defect to me. This problem is corrected in Figure 4d. My suggestion is to eliminate Figure 4a and to replace it with the current Figure 4d as Figure 4a followed by the current Figures 4b and 4c. Additional rewording of the legend might be necessary to accommodate this change.
1) Line 10. Delete “generally.”
2) Line 13. Write as “by an ion….”
3) Line 17. Replace “reversing” with “reversible.” Also on line 45.
4) Line 28. Delete “commonly.”
5) Line 30. Delete “Gram-negative.” Gram-positive bacteria also have external flagella.
6) Line 42. Rewrite as “…a variable number of stator units.”
7) Line 50. Start new paragraph.
8) Line 52, Replace “a” with “the.”
9) Lines 61-62. Rewrite as “…the basal body, although they are not shown….”
10) Lines 66-67. Rewrite as “…networks. E. coli and Salmonella….”
11) Line 68. Replace “stimuli” with “environments.”
12) Line 70. Change “environments” to “environment.”
13) Line 73. Replace “the” with “a.”
14) Line 98. Insert comma after “structure.”
15) Line 105. Replace “takes” either with “has” or with “takes on.” Also, delete “that.”
16) Line 114. The “wherRe: FliP_mutanteas FliMC” makes no sense. Revise.
17) Line 117. If the authors are Shohei Ohtani fans and really want to say “Dodger blue,” Dodger should be capitalized. Otherwise, choose another shade of blue. Also, it should be “FlgH subunits are colored.”
18) Line 118. I think it should be “N-terminal disordered chain of FlgI binds to FlgH.”
19) Line 133-134. Rewrite as “As a result, 23 RBM2 domains of the 34 FliF proteins form the inner core of the M-ring, while the….”
20) Lines 142-143. Delete “in E. coli and Salmonella.”
21) Lines 146-147. Delete “between.”
22) Line 152. Insert “for” after “as well as.”
23) Line 162. Replace “on” with “within.”
24) Line 172. Should “FiMN” really be “FliNN”?
25) Lines 176-177. Divide into two sentences. “…lipoylation. It forms….”
26) Lines 181-182. Rewrite as “around the negatively charged surface of the rod [65].”
27) Line 187. Rewrite as “…of the rod, which acts as the drive shaft…..”
28) Lines 195-196. Rewrite as “…strongly negative because of three negatively charged residues in FlgG, Asp….”
29) Line 198. Start new paragraph with “Because of the difference in the cell envelope of….”
30) Line 225. Rewrite as “Domains D0, Dc, and D1 of FlgE are structurally similar to those domains in FlgG….”
31) Lines 229-230. Replace “flexible against bending” with “flexible for bending.”
32) Line 231. Delete “in a way similar to the rod.”
33) Line 238. Delete “motor.”
34) Line 262. Rewrite as “Flagellin is the building block….”
35) Line 265. Replace “…in a way similar to” with “as in.”
36) Line 269. Replace “adopter” with “adapter.”
37) Line 270. Replace “deficient” with “missing.”
38) Line 271.Delete “so.”
39) Line 284. Begin new paragraph with “Polymorphic transformation….”
40) Line 298. Insert comma between “species” and “whereas.”
41) Line 301. Replace “autonomously regulated” with “stochastically switched.”
42) Line 303. Rewrite as “This alternating expression….”
43) Line 305. Start new paragraph “The motility of Salmonella….” Delete “Interestingly.”
44) Line 318. Replace “with” with “of.”
45) Line 321. Insert “the” to read “uses the proton motive force….”
46) Line 323. Rewrite as “through an interaction.”
47) Line 328. Rewrite as “…platform achieve the proper order of flagellar protein export….”
48) Line 332. Rewrite as “allows proteins of the axial structure of the flagellum to pass….”
49) Line 335. The “assemble” should be “assembles.”
50) Line 350. Set off with commas: “….recovery, called “resurrection,” indicates….”
51) Lines 351-352. Rewrite as “…motor rotation. These fluctuations are caused by….”
52) Lines 374-378. Rewrite as “Because each stator unit is stabilized around the rotor via a tight association between the peptidoglycan-binding (PGB) domain of MotB and the PG layer, the dependence of stator assembly on the coupling ion probably involves ion-dependent structural changes in the stator unit that open the ion channel (Figure 4a) [146–148].
53) Line 381. Replace “against” with “on.”
54) Lines 385-387. Rewrite as “…after attaching a bead to a stub of a “sticky” flagellar filament and showed that abrupt increases in load cause stepwise increases in motor speed [150].”
55) Line 391. Rewrite as “…there is typically only one stator unit….”
56) Line 405. Begin new paragraph with “Measurements of stator kinetics….”
57) Line 407. Replace “averaged” with “average,” unless averaged is something specific.
58) Line 411. Delete “Interestingly.”
59) Line 419. Replace “nonrotation” with “stationary” or “non-rotating.”
60) Line 424. Delete “as well as the unbound state.” It makes no sense. Is this the “hidden state” of Wadwha et al.?”
61) Lines 428-429. Very confusing as written. I suggest “Wadhwa et al. have proposed a model in which stator binding is promoted as motor speed increases, resulting in an increase in the number of functional stator units around a rotor [153].
62) Lines 435-436. Delete “presumably.”
63) Line 437. Delete “the.”
64) Line 438, Insert comma after “rotor.”
65) Lines 439-440. Not clear what “but the peaked binding rate against the number of installed stator units” means Perhaps it should be “but the binding rate peaked at an intermediate number of installed stator units.”
66) Lines 442-444. Rewrite as “One of the parameters is that at increased motor speed there is an increased probability of collision between an unbound stator and the rotor; the other is that there is a decreased probability of forming a permanent binding of a stator unit because of its shorter contact time with the more-rapidly spinning rotor.”
67) Line 449. Should be “double-exponential.”
68) Line 450 and 452. Delete “the.”
69) Line 457. Delete “largely.”
70) Lines 455-461. Start new paragraph and rewrite for clarity. “The kinetics of stator remodeling is best described by a four-state mechanism. Noting the load dependence of the stator-docking lifetime, there is a loose-binding state, in which the stretch of the stator anchoring region is limited because of low torque, and a tight-binding state, where the stator anchoring region stretches in response to high torque. There are also two unbound states, one being free diffusion in the membrane and the other being the hidden state proposed by Shi et al., in which the stator unit is still bound to PG at the periphery of the rotor but is not engaged with FliG (Figure 4a).” [Fig. 4a in the suggested new order.]
71) Line 461. Delete “the.”
72) Line 464. Delete the second “the.”
73) Line 773. Rewrite as “Based on this mechanism, theorists….”
74) Lines 481-482. Rewrite as “A model for CCW/CW bidirectional rotor rotation based on unidirectional CW stator rotation (viewed from the cytoplasmic membrane looking down onto the FliG ring; lower panels.)
75) Line 486. Rewrite as “…near-atomic resolution. These structures have completely revised models for the mechanism of rotor rotation.”
76) Line 494. Begin new paragraph with “The five MotA….”
77) Line 495. Should be “coiled-coil.”
78) Lines 496-501. Rewrite as “Limited proteolysis assays showed that the mutation causing the D32N residue substitution in E. coli MotB, which mimics the protonation of MotB-Asp32, causes a large conformation of the cytoplasmic domain of MotA (MotAc) [164].
The authors proposed that this change reflects a power-stroke mechanism for the stator. In contrast, the cryo-EM structure of the MotA5-MotB2 ring complex from C. jejuni reveals that the D22A change does not induce a large conformational change in MotAc [ref.?] (D22 in C jejuni corresponds to D32 in E. coli.
79) Line 501. Start new paragraph with “The following estimate suggests….”
80) Line 503 ff. A bit hard to follow. I suggest starting a new paragraph and rewriting thus. “The rotor ring, with a diameter of ~450 Å and a circumference of ~1420 Å, is formed by 34 FliG subunits [25]. Therefore, adjacent FliG subunits are about 1420/34 = ~42 Å arc length apart. That should be the elementary step length of the motor. A root-mean-square-deviation (RMSD) scoring the structural difference in Cα atoms between the deprotonated and protonated states of the MotA5-MotB2 ring complex is only 0.297 Å. Thus, there must be very considerable extra movement of MotAc coupled with H+ translocation through the ion channel.
81) Line 508 ff. Suggest starting new paragraph and rewriting as follows. “The diameter of the MotA5-MotB2 ring structure is ~75 Å and the circumference is ~ 230 Å. In a 72° rotation of the MotA ring around of the MotB dimer, it will move through an arc of ~46 Å. Thus, the influx of H+ through the H+ channel induces rotation of the MotA5 ring of the proper size, given the inaccuracy of the precise measurements of distance, of just the right magnitude to propel a 42 Å rotation of the FliG rotor ring. Simlar H+-coupled structural changes in the stator complex are also observed in the spirochete Borrelia burgdorferi [165]. The rotational switching mechanism proposed for the B. burgdorferi flagellar motor employs a similar mechanism of stator rotation (discussed below).
82) Line 517. Replace “for” with “to power.”
83) Line 519. Replace “the” with “a.”
84) Line 522. Start new paragraph with “FliGc is divided….”
85) Line 526. Replace “of with “in.”
86) Line 536. Replace “pathway” with “pathways.”
87) Line 541. Replace “on” with “into.”
88) Line 542.Insert comma between “(CCW)” and “whereas.”
89) Line 545. Start new paragraph with “Johnson.”
90) Line 550. Insert comma after “inside.”
91) Line 552. Replace “described” with “describes.”
92) Line 555. Inset “the” before “symmetry.”
93) Line 338. Delete comma.
94) Line 561. Start new paragraph with “Thus, although….”
95) Lines 571-572. Start new sentence. “…in vitro. (Reconstitution…complexes.)”
Reviewer 2 Report
Comments and Suggestions for Authors
This review paper provides a detailed review of the molecular structure of flagellar complex and its function and biological relevance. I have a few concerns to be addressed by the authors before supporting its publication.
The Introduction section delves into technical details rather than providing an overview of the impact of this work. Consequently, it contains unnecessary redundancy with later sections. For instance, in the first paragraph on page 2, “… serves as an initial template for flagellar assembly” almost duplicates the sentence “… but also as an initial template for flagellar assembly” in the second paragraph on page 4. This repetition also causes an abrupt transition between the last two paragraphs in the Introduction section.
The discussion of the interplay between molecular structure and polymorphic transformation appears to be an exception only for peritrichous bacteria, such as E. coli. It does not apply to many monotrichous species common in other bacterial types. For instance, the reversal of the motor does not cause any switch in polymorphic transformation for marine bacteria like Vibrio alginolyticus or freshwater bacteria like Caulobacter crescentus. As a review paper, this flagellar structure should be discussed in a broader manner.
In the caption of Figure 2, the subfigure is mislabeled. “(b) C{alpha} ribbon…” should be “(c) C{alpha} ribbon…”
In the last paragraph on page 9, “… share flows” should be “… shear flows.”
Reviewer 3 Report
Comments and Suggestions for Authors
The authors review structure-function relationships in the flagella of Escherichia coli and Salmonella enterica, which are the paradigm for bacterial flagella. A number of papers that analyze the flagellar structures by cryo-electron microscopy have been published recently, and the authors do an excellent job in summarizing the results of these papers. The review is clearly written and is of broad interest.
Specific comments:
1. On line 176 the authors indicate that FlgH has “an N-terminal common motif (LTGC) for lipoylation”. The motif that is recognized by the enzymes that acylate the conserved cysteine (Lgt and Lnt) and cleave the signal peptide (Lsp) is not that specific. It does not seem that important to indicate the motif, and I suggest shortening the sentence to read, “FlgH is a lipoprotein that forms the L-ring within the outer membrane [61].”
2. On lines 188-189, the authors indicate that “the rod and the inner surface of the LP-ring are very smooth”. I am not sure what is meant by this statement. Are the surfaces of these structures significantly smoother than those of other cellular structures?
3. It is understandable why Bacillus subtilis lacks the L-ring since it does not possess an outer membrane. It is not obvious why B. subtilis lacks the P-ring though. Does the peptidoglycan layer function as the bushing for the flagellar rod in B. subtilis? Is it known that other Gram-negative bacteria lack the P-ring or is B. subtilis possibly unique in this regard? It would be interesting to have the authors discuss this more, and even provide some speculation if needed.
4. The organization of the outer membrane around the L-ring is unusual as it is not a typical lipid bilayer as discussed in reference 28 of the manuscript. The authors do not discuss this, but it seems that it is worth discussing.
5. There have been a couple of recent papers on the stator-associated protein FliL, yet there is no discussion of FliL in the manuscript. This seems a major oversight. Some relevant references on FliL are:
Tachiyama S, Chan KL, Liu X, Hathroubi S, Peterson B, Khan MF, Ottemann KM, Liu J, Roujeinikova A. The flagellar motor protein FliL forms a scaffold of circumferentially positioned rings required for stator activation. Proc Natl Acad Sci U S A. 2022 Jan 25;119(4):e2118401119. doi: 10.1073/pnas.2118401119
Takekawa N, Isumi M, Terashima H, Zhu SNishino Y, Sakuma MKojima S, Homma M, Imada K. 2019. Structure of Vibrio FliL, a New Stomatin-like Protein That Assists the Bacterial Flagellar Motor Function. mBio 10:10.1128/mbio.00292-19
Partridge JD, Harshey RM. Flagellar protein FliL: A many-splendored thing. Mol Microbiol. 2024 Oct;122(4):447-454. doi: 10.1111/mmi.15301
Liu X, Roujeinikova A, Ottemann KM. FliL Functions in Diverse Microbes to Negatively Modulate Motor Output via Its N-Terminal Region. mBio. 2023 Apr 25;14(2):e0028323. doi: 10.1128/mbio.00283-23. Epub 2023 Feb 28. Erratum in: mBio. 2023 Dec 19;14(6):e0239623. doi: 10.1128/mbio.02396-23
Minor comments:
1. In line 300, fljB should be italicized.
Reviewer 4 Report
Comments and Suggestions for Authors
The review paper by Nakamura and Minamino describes the present-day findings on structure and dynamics of the bacterial rotary flagellar motor. The authors provide a concise and yet comprehensive review of the latest developments in understanding the structure and mechanisms of bacterial flagellar motor. Many of the studies reviewed by the authors are their own original work published in highly reputable journals, which undoubtedly confirms their expertise. Undoubtedly, the review will be very useful for those interested in both the mechanisms of bacterial flagellar motor functioning and general biology. The publication is well-justified.
A few points require attention before publication.
In the abstract, the authors state that the review article "describes the current understanding of the structure and dynamics of the proton-driven flagellar motor complex of Escherichia coli and Salmonella enteric serovar Typhimurium" (lines 21-23). However, the review covers data for a wider range of bacteria, especially in Section 4. It seems that the authors should make changes to the abstract to reflect this. Additionally, the review discusses Na+-driven bacterial motors, which have mechanisms that are not significantly different from those of proton-driven motors. Therefore, the word "proton-driven" may be removed from the title.
At line 24, the keyword "cryo-electron microscope" should be changed to "cryo-electron microscopy".
Figure 3, lines 220-226: Please provide the full names of the proteins FlgG, FlgE, and FliC.
In Section 2.6, I recommend that the authors include a spatial structure of a region of the flagellin filament similar to that shown in Figure 3 to illustrate this section.
Lines 290-294: It would be helpful if the authors could describe the results presented in [110] more thoroughly, such as the differences in the conformational states of the 11 protofilaments between normal and curly filaments.
